# HolStep: A Machine Learning Dataset for Higher-order Logic Theorem Proving

**Cezary Kaliszyk**
University of Innsbruck
`cezary.kaliszyk@uibk.ac.at`

**François Chollet, Christian Szegedy**
Google Research
`{fchollet,szegedy}@google.com`

## ABSTRACT

Large computer-understandable proofs consist of millions of intermediate logical steps. The vast majority of such steps originate from manually selected and manually guided heuristics applied to intermediate goals. So far, machine learning has generally not been used to filter or generate these steps. In this paper, we introduce a new dataset based on Higher-Order Logic (HOL) proofs, for the purpose of developing new machine learning-based theorem-proving strategies. We make this dataset publicly available under the BSD license. We propose various machine learning tasks that can be performed on this dataset, and discuss their significance for theorem proving. We also benchmark a set of simple baseline machine learning models suited for the tasks (including logistic regression, convolutional neural networks and recurrent neural networks). The results of our baseline models show the promise of applying machine learning to HOL theorem proving.

## 1 INTRODUCTION

As the usability of interactive theorem proving (ITP) systems (Harrison et al., 2014) grows, its use becomes a more common way of establishing the correctness of software as well as mathematical proofs. Today, ITPs are used for software certification projects ranging from compilers (Leroy, 2009) and operating system components (Chen et al., 2016; Klein et al., 2014), to establishing the absolute correctness of large proofs in mathematics such as the Kepler conjecture (Hales et al., 2015) and the Feit-Thomson Theorem (Gonthier et al., 2013).

For results of such significance to be possible, the theorem libraries of these ITPs must contain all necessary basic mathematical properties, accompanied with formal proofs. This means that the size of many ITP libraries can be measured in dozens of thousands of theorems (Grabowski et al., 2010; Blanchette et al., 2015) and billions of individual proof steps. While the general direction of the proofs is specified by humans (by providing the goal to prove, specifying intermediate steps, or applying certain automated tactics), the majority of such proof steps are actually found by automated reasoning-based proof search (Kaliszyk & Urban, 2015b), with very little application of machine learning techniques so far.

At the same time, fast progress has been unfolding in machine learning applied to tasks that involve logical inference, such as natural language question answering (Sukhbaatar et al., 2015), knowledge base completion (Socher et al., 2013a), automated translation (Wu et al., 2016), and premise selection in the context of theorem proving (Alemi et al., 2016). Deep learning in particular has proven to be a powerful tool for embedding semantic meaning and logical relationships into geometric spaces, specifically via models such as convolutional neural networks, recurrent neural networks, and tree-recursive neural networks. These advances strongly suggest that deep learning may have become mature enough to yield significant advances in automated theorem proving. Remarkably, it has recently become possible to build a system, AlphaGo (Silver et al., 2016), blending classical AI techniques such as Monte-Carlo tree search and modern deep learning techniques, capable of playing the game of Go at super-human levels. We should note that theorem proving and Go playing are conceptually related, since both consist in searching for specific nodes in trees of states with extremely large arity and relatively large depth, which involves node evaluation decision (how valuable is this state?) and policy decisions (which node should be expanded next?). The success of AlphaGo can thus serve as encouragement on the road to building deep learning-augmented theorem

provers that would blend classical techniques developed over the past few decades with the latest machine learning advances.

Fast progress in specific machine learning verticals has occasionally been achieved thanks to the release of specialized datasets (often with associated competitions, e.g. the ImageNet dataset for large-scale image classification (Deng et al., 2009)) serving as an experimental testbed and public benchmark of current progress, thus focusing the efforts of the research community. We hope that releasing a theorem proving dataset suited for specific machine learning tasks can serve the same purpose in the vertical of applying machine learning to theorem proving.

## 1.1 CONTRIBUTION AND OVERVIEW

First, we develop a dataset for machine learning based on the proof steps used in a large interactive proof section 2. We focus on the HOL Light (Harrison, 2009) ITP, its multivariate analysis library (Harrison, 2013), as well as the formal proof of the Kepler conjecture (Hales et al., 2010). These formalizations constitute a diverse proof dataset containing basic mathematics, analysis, trigonometry, as well as reasoning about data structures such as graphs. Furthermore these formal proof developments have been used as benchmarks for automated reasoning techniques (Kaliszyk & Urban, 2014).

The dataset consists of 2,013,046 training examples and 196,030 testing examples that originate from 11,400 proofs. Precisely half of the examples are statements that were useful in the currently proven conjectures and half are steps that have been derived either manually or as part of the automated proof search but were not necessary in the final proofs. The dataset contains only proofs of non-trivial theorems, that also do not focus on computation but rather on actual theorem proving. For each proof, the conjecture that is being proven as well as its dependencies (axioms) and may be exploited in machine learning tasks. Furthermore, for each statement both its human-readable (pretty-printed) statement and a tokenization designed to make machine learning tasks more manageable are included.

Next, in section 3 we discuss the proof step classification tasks that can be attempted using the dataset, and we discuss the usefulness of these tasks in interactive and automated theorem proving. These tasks include unconditioned classification (without access to conjectures and dependencies) and conjecture-conditioned classification (with access to the conjecture) of proof steps as being useful or not in a proof. We outline the use of such classification capabilities for search space pruning and internal guidance, as well as for generation of intermediate steps or possible new lemma statements.

Finally, in section 4 we propose three baseline models for the proof step classification tasks, and we experimentally evaluate the models on the data in section 5. The models considered include both a relatively simple regression model, as well as deep learning models based on convolutional and recurrent neural networks.

## 1.2 RELATED WORK

The use of machine learning in interactive and automated theorem proving has so far focused on three tasks: premise selection, strategy selection, and internal guidance. We shortly explain these.

Given a large library of proven facts and a user given conjecture, the multi-label classification problem of selecting the facts that are most likely to lead a successful proof of the conjecture has been usually called *relevance filtering* or *premise selection* (Alama et al., 2014). This is crucial for the efficiency of modern automation techniques for ITPs (Blanchette et al., 2016), which today can usually solve 40–50% of the conjectures in theorem proving libraries. Similarly most competitive ATPs today (Sutcliffe, 2016) implement the SInE classifier (Hoder & Voronkov, 2011).

A second theorem proving task where machine learning has been of importance is *strategy selection*. With the development of automated theorem provers came many parameters that control their execution. In fact, modern ATPs, such as E (Schulz, 2013) and Vampire (Kovács & Voronkov, 2013), include complete strategy description languages that allow a user to specify the orderings, weighting functions, literal selection strategies, etc. Rather than optimizing the search strategy globally, one

can choose the strategy based on the currently considered problem. For this some frameworks use machine learning (Bridge et al., 2014; Kühlwein & Urban, 2015).

Finally, an automated theorem prover may use machine learning for choosing the actual inference steps. It has been shown to significantly reduce the proof search in first-order tableaux by the selection of extension steps to use (Urban et al., 2011), and has been also successfully applied in monomorphic higher-order logic proving (Färber & Brown, 2016). Data/proof mining has also been applied on the level of interactive theorem proving tactics (Duncan, 2007) to extract and reuse repeating patterns.

## 2 DATASET EXTRACTION

We focus on the HOL Light theorem prover for two reasons. First, it follows the LCF approach[1]). This means that complicated inferences are reduced to the most primitive ones and the data extraction related modifications can be restricted the primitive inferences and it is relatively easy to extract proof steps at an arbitrary selected level of granularity. Second, HOL Light implements higher-order logic (Church, 1940) as its foundation, which on the one hand is powerful enough to encode most of today's formal proofs, and on the other hand allows for an easy integration of many powerful automation mechanisms (Baader & Nipkow, 1998; Paulson, 1999).

When selecting the theorems to record, we choose an intermediate approach between HOL Light ProofRecording (Obua & Skalberg, 2006) and the HOL/Import one (Kaliszyk & Krauss, 2013). The theorems that are derived by most common proof functions are extracted by patching these functions like in the former approach, and the remaining theorems are extracted from the underlying OCaml programming language interpreter. In certain cases decision procedures derive theorems to be reused in subsequent invocations. We detect such values by looking at theorems used across proof blocks and avoid extracting such reused unrelated subproofs.

All kernel-level inferences are recorded together with their respective arguments in a trace file. The trace is processed offline to extract the dependencies of the facts, detect used proof boundaries, mark the used and unused steps, and mark the training and testing examples. Only proofs that have sufficiently many used and unused steps are considered useful for the dataset. The annotated proof trace is processed again by a HOL kernel saving the actual training and testing examples originating from non-trivial reasoning steps. Training and testing examples are grouped by proof: for each proof the conjecture (statement that is finally proved), the dependencies of the theorem are constant, and a list of used and not used intermediate statements is provided. This means that the conjectures used in the training and testing sets are normally disjoint.

For each statement, whether it is the conjecture, a proof dependency, or an intermediate statement, both a fully parenthesised HOL Light human-like printout is provided, as well as a predefined tokenization. The standard HOL Light printer uses parentheses and operator priorities to make its notations somewhat similar to textbook-style mathematics, while at the same time preserving the complete unambiguity of the order of applications (this is particularly visible for associative operators). The tokenization that we propose attempts to reduce the number of parentheses. To do this we compute the maximum number of arguments that each symbol needs to be applied to, and only mark partial application. This means that fully applied functions (more than 90% of the applications) do not require neither application operators nor parentheses. Top-level universal quantifications are eliminated, bound variables are represented by their de Bruijn indices (the distance from the corresponding abstraction in the parse tree of the term) and free variables are renamed canonically. Since the Hindley-Milner type inference Hindley (1969) mechanisms will be sufficient to reconstruct the most-general types of the expressions well enough for automated-reasoning techniques Kaliszyk et al. (2015) we erase all type information. Table 1 presents some dataset statistics. The dataset, the description of the used format, the scripts used to generate it and baseline models code are available:

http://cl-informatik.uibk.ac.at/cek/holstep/

---

[1]LCF approach is a software architecture for implementing theorem provers which uses a strongly typed programming language with abstract datatypes (such as OCaml in the case of HOL Light) to separate the small trusted core, called the kernel, which verifies the primitive inferences from user code which allows the user to arbitrarily extend the system in a safe manner. For more details see (Gordon et al., 1979).

|                   | Train   | Test   | Positive | Negative |
|-------------------|---------|--------|----------|----------|
| Examples          | 2013046 | 196030 | 1104538  | 1104538  |
| Avg. length       | 503.18  | 440.20 | 535.52   | 459.66   |
| Avg. tokens       | 87.01   | 80.62  | 95.48    | 77.40    |
| Conjectures       | 9999    | 1411   | -        | -        |
| Avg. dependencies | 29.58   | 22.82  | -        | -        |

Table 1: HolStep dataset statistics

## 3 MACHINE LEARNING TASKS

### 3.1 TASKS DESCRIPTION

This dataset makes possible several tasks well-suited for machine learning most of which are highly relevant for theorem proving:

- Predicting whether a statement is useful in the proof of a given conjecture;

- Predicting the dependencies of a proof statement (premise selection);

- Predicting whether a statement is an important one (human named);

- Predicting which conjecture a particular intermediate statement originates from;

- Predicting the name given to a statement;

- Generating intermediate statements useful in the proof of a given conjecture;

- Generating the conjecture the current proof will lead to.

In what follows we focus on the first task: classifying proof step statements as being useful or not in the context of a given proof. This task may be further specialized into two different tasks:

- Unconditioned classification of proof steps: determining how likely a given proof is to be useful for the proof it occurred in, based solely on the content of statement (i.e. by only providing the model with the step statement itself, absent any context).

- Conditioned classification of proof steps: determining how likely a given proof is to be useful for the proof it occurred in, with "conditioning" on the conjecture statement that the proof was aiming to attain, i.e. by providing the model with both the step statement and the conjecture statement).

In the dataset, for every proof we provide the same number of useful and non-useful steps. As such, the proof step classification problem is a balanced two-class classification problem, where a random baseline would yield an accuracy of 0.5.

### 3.2 RELEVANCE TO INTERACTIVE AND AUTOMATED THEOREM PROVING

In the interaction with an interactive theorem prover, the tasks that require most human time are: the search for good intermediate steps; the search for automation techniques able to justify the individual steps, and searching theorem proving libraries for the necessary simpler facts. These three problems directly correspond to the machine learning tasks proposed in the previous subsection. Being able to predict the usefulness of a statement will significantly improve many automation techniques. The generation of good intermediate lemmas or intermediate steps can improve level of granularity of the proof steps. Understanding the correspondence between statements and their names can allow users to search for statements in the libraries more efficiently (Aspinall & Kaliszyk, 2016). Premise selection and filtering are already used in many theorem proving systems, and generation of succeeding steps corresponds to conjecturing and theory exploration.

Figure 1: Unconditioned classification model architectures.

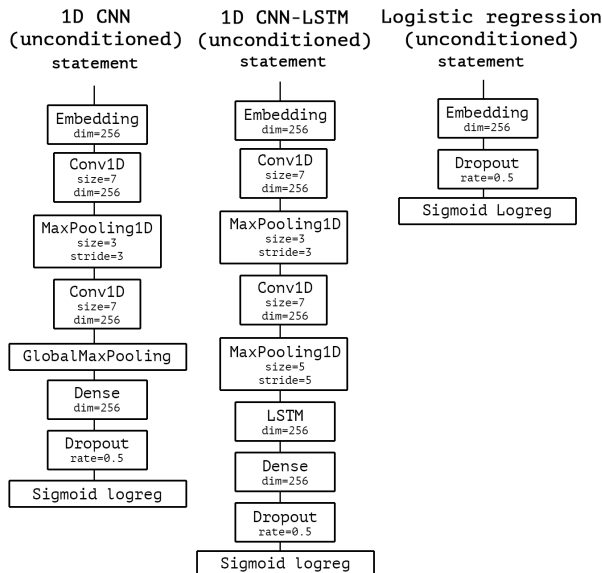

# 4 BASELINE MODELS

For each task (conditioned and unconditioned classification), we propose three different deep learning architectures, meant to provide a baseline for the classification performance that can be achieved on this dataset. Our models cover a range of architecture features (from convolutional networks to recurrent networks), aiming at probing what characteristics of the data are the most helpful for usefulness classification.

Our models are implemented in TensorFlow (Abadi et al., 2015) using the Keras framework (Chollet, 2015). Each model was trained on a single Nvidia K80 GPU. Training only takes a few hours per model, which makes running these experiments accessible to most people (they could even be run on a laptop CPU). We are releasing all of our benchmark code as open-source software [2] so as to allow others to reproduce our results and improve upon our models.

## 4.1 UNCONDITIONED CLASSIFICATION MODELS

Our three models for this task are as follow:

- Logistic regression on top of learned token embeddings. This minimal model aims to determine to which extent simple differences between token distribution between useful and non-useful statements can be used to distinguish them. It provides an absolute floor on the performance achievable on this task.
- 2-layer 1D convolutional neural network (CNN) with global maxpooling for sequence reduction. This model aims to determine the importance of local patterns of tokens.
- 2-layer 1D CNN with LSTM (Hochreiter & Schmidhuber, 1997) sequence reduction. This model aims to determine the importance of order in the features sequences.

See figure 1 for a layer-by-layer description of these models.

## 4.2 CONDITIONED CLASSIFICATION MODELS

For this task, we use versions of the above models that have two siamese branches (identical branches with shared weights), with one branch processing the proof step statement being considered, and the

---

[2]https://github.com/tensorflow/deepmath/tree/master/holstep_baselines

Figure 2: Conditioned classification model architectures.

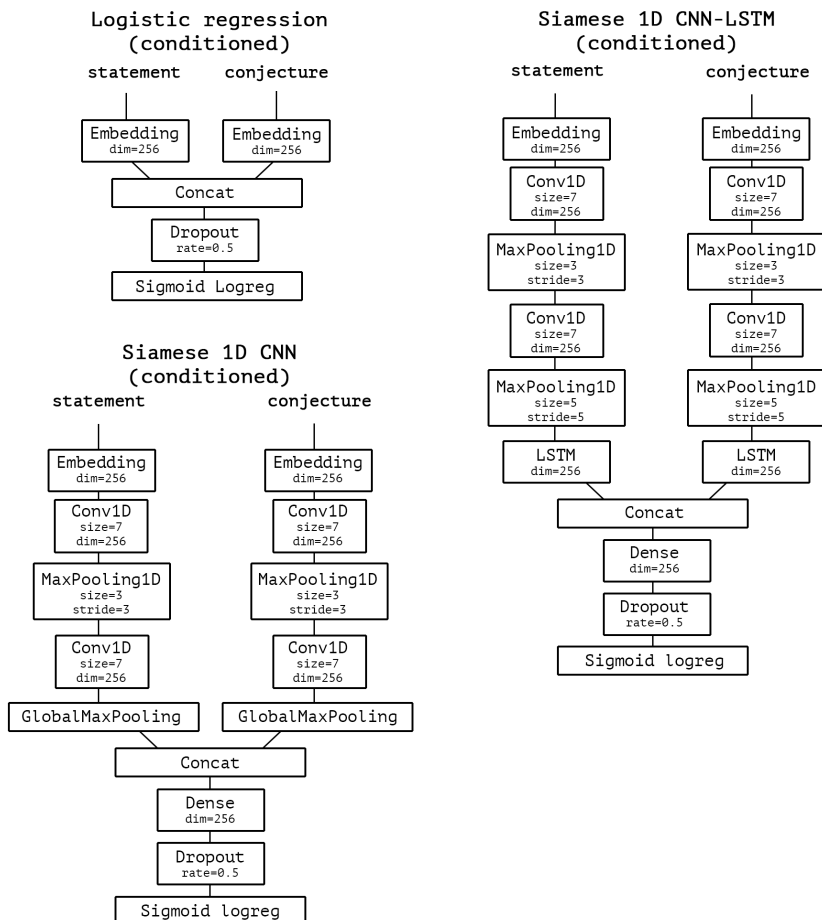

other branch processing the conjecture. Each branch outputs an embedding; these two embeddings (step embedding and conjecture embedding) are then concatenated and the classified by a fully-connected network. See figure 2 for a layer-by-layer description of these models.

## 4.3 INPUT STATEMENTS ENCODING

It should be noted that all of our models start with an Embedding layer, mapping tokens or characters in the statements to dense vectors in a low-dimensional space. We consider two possible encodings for presenting the input statements (proof steps and conjectures) to the Embedding layers of our models:

- Character-level encoding of the human-readable versions of the statements, where each character (out of a set of 86 unique characters) in the pretty-printed statements is mapped to a 256-dimensional dense vector. This encoding yields longer statements (training statements are 308 character long on average).

- Token-level encoding of the versions of the statements rendered with our proposed high-level tokenization scheme. This encoding yields shorter statements (training statements are 60 token long on average), while considerably increasing the size of set of unique tokens (1993 total tokens in the training set).

Table 2: HolStep proof step classification accuracy without conditioning

| | Logistic regression | 1D CNN | 1D CNN-LSTM |
|---|---|---|---|
| **Accuracy with char input** | 0.71 | 0.82 | **0.83** |
| **Accuracy with token input** | 0.71 | **0.83** | 0.77 |

Table 3: HolStep proof step classification accuracy with conditioning

| | Logistic regression | Siamese 1D CNN | Siamese 1D CNN-LSTM |
|---|---|---|---|
| **Accuracy with char input** | 0.71 | 0.81 | **0.83** |
| **Accuracy with token input** | 0.71 | 0.82 | 0.77 |

## 5 RESULTS

Experimental results are presented in tables 2 and 3, as well as figs. 3 to 6.

### 5.1 INFLUENCE OF MODEL ARCHITECTURE

Our unconditioned logistic regression model yields an accuracy of 71%, both with character encoding and token encoding (tables 2 and 3). This demonstrates that differences in token or character distributions between useful and non-useful steps alone, absent any context, is sufficient for discriminating between useful and non-useful statements to a reasonable extent. This also demonstrates that the token encoding is not fundamentally more informative than raw character-level statements.

Additionally, our unconditioned 1D CNN model yields an accuracy of 82% to 83%, both with character encoding and token encoding (tables 2 and 3). This demonstrates that patterns of characters or patterns of tokens are considerably more informative than single tokens for the purpose of usefulness classification.

Finally, our unconditioned convolutional-recurrent model does not improve upon the results of the 1D CNN, which indicates that our models are not able to meaningfully leverage order in the feature sequences into which the statements are encoded.

### 5.2 INFLUENCE OF INPUT ENCODING

For the logistic regression model and the 2-layer 1D CNN model, the choice of input encoding seems to have little impact. For the convolutional-recurrent model, the use of the high-level tokenization seems to cause a large decrease in model performance (figs. 4 and 6). This may be due to the fact that token encoding yields shorter sequences, making the use of a LSTM less relevant.

### 5.3 INFLUENCE OF CONDITIONING ON THE CONJECTURE

None of our conditioned models appear to be able to improve upon the unconditioned models, which indicates that our architectures are not able to leverage the information provided by the conjecture. The presence of the conditioning does however impact the training profile of our models, in particular by making the 1D CNN model converge faster and overfit significantly quicker (figs. 5 and 6).

## 6 CONCLUSIONS

Our baseline deep learning models, albeit fairly weak, are still able to predict statement usefulness with a remarkably high accuracy. Such methods already help first-order automated provers (Kaliszyk & Urban, 2015a) and as the branching factor is higher in HOL the predictions are valuable for a number of practical proving applications. This includes making tableaux-based (Paulson, 1999) and superposition-based (Hurd, 2003) internal ITP proof search significantly more efficient in turn

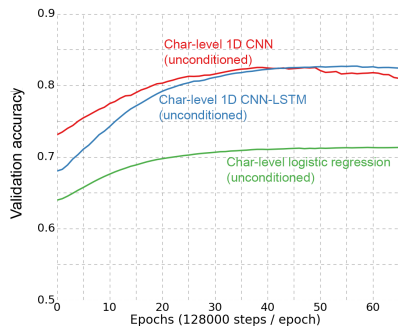

Figure 3: Training profile of the three unconditioned baseline models with character input.

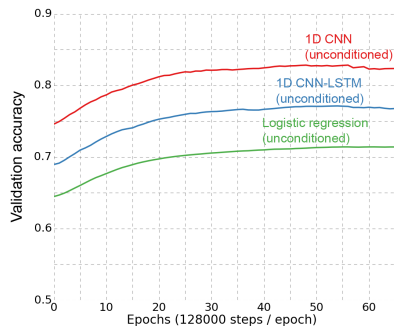

Figure 4: Training profile of the three unconditioned baseline models with token input.

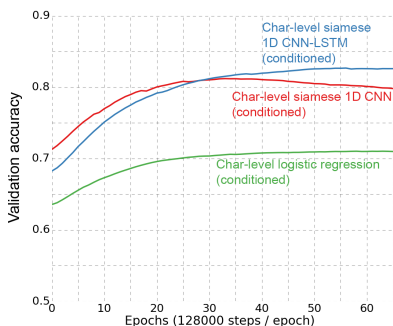

Figure 5: Training profile of the three conditioned baseline models with character input.

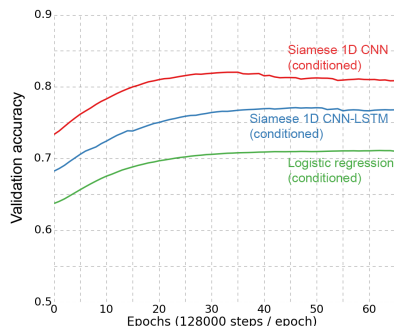

Figure 6: Training profile of the three conditioned baseline models with token input.

making formalization easier. However, our models do not appear to be able to leverage order in the input sequences, nor conditioning on the conjectures. This is due to the fact that these models are not doing any form of logical reasoning on their input statements; rather they are doing simple pattern matching at the level of n-grams of characters or tokens. This shows the need to focus future efforts on different models that can do *reasoning*, or alternatively, on systems that blend explicit reasoning (e.g. graph search) with deep learning-based feature learning. A potential new direction would be to leverage the graph structure of HOL statements using e.g. Recursive Neural Tensor Networks (Socher et al., 2013a;b) or other graph-based recursive architectures.

## 6.1 FUTURE WORK

The dataset focuses on one interactive theorem prover. It would be interesting if the proposed techniques generalize, primarily across ITPs that use the same foundational logic, for example using OpenTheory (Hurd, 2011), and secondarily across fundamentally different ITPs or even ATPs. A significant part of the unused steps originates from trying to fulfill the conditions for rewriting and from calls to intuitionistic tableaux. The main focus is however on the human found proofs so the trained predictions may to an extent mimic the bias on the usefulness in the human proofs. As ATPs are at the moment very week in comparison with human intuition improving this even for the many proofs humans do not find difficult would be an important gain.

Finally, two of the proposed task for the dataset have been premise selection and intermediate sentence generation. It would be interesting to define more ATP-based ways to evaluate the selected premises, as well as to evaluate generated sentences (Kaliszyk et al., 2015). The set is a relatively large one when it comes to proof step classification, however the number of available premises makes the set a medium-sized set for premise selection in comparison with those of the Mizar Mathematical Library or the seL4 development.

ACKNOWLEDGEMENTS

The first author was partly supported by the ERC starting grant 714034.

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
