# Peer review of "HolStep: A Machine Learning Dataset for Higher-order Logic Theorem Proving"

_ICLR 2017 — accepted_

[Official Review · AnonReviewer1 · rating 8 · confidence 3 · 16 Dec 2016]
**No Title**

The authors present a dataset extraction method, dataset and first interesting results for machine-learning supported higher order logic theorem proving. The experimental results are impressively good for a first baseline and with an accuracy higher than 0.83 in relevance classification a lot better than chance, and encourage future research in this direction. The paper is well-written in terms of presentation and argumentation and leaves little room for criticism. The related work seems to be well-covered, though I have to note that I am not an expert for automated theorem proving.

[Official Review · AnonReviewer2 · rating 7 · confidence 3 · 16 Dec 2016]
**Great start, interested in future work**

The authors describe a dataset of proof steps in higher order logic derived from a set of proven theorems. The success of methods like AlphaGo suggests that for hard combinatorial style problems, having a curated set of expert data (in this case the sequence of subproofs) is a good launching point for possibly super-human performance. Super-human ATPs are clearly extremely valuable. Although relatively smaller than the original Go datasets, this dataset seems to be a great first step. Unfortunately, the ATP and HOL aspect of this work is not my area of expertise. I can't comment on the quality of this aspect.

It would be great to see future work scale up the baselines and integrate the networks into state of the art ATPs. The capacity of deep learning methods to scale and take advantage of larger datasets means there's a possibility of an iterative approach to improving ATPs: as the ATPs get stronger they may generate more data in the form of new theorems. This may be a long way off, but the possibility is exciting.

[Official Review · AnonReviewer3 · rating 6 · confidence 3 · 23 Dec 2016 (modified: 22 Jan 2017)]
**Interesting dataset, reasonable first steps**

Use of ML in ITP is an interesting direction of research. Authors consider the problem of predicting whether a given statement would be useful in a proof of a conjecture or not. This is posed as a binary classification task and authors propose a dataset and some deep learning based baselines. 

I am not an expert on ITP or theorem proving, so I will present a review from more of a ML perspective. I feel one of the goals of the paper should be to present the problem to a ML audience in a way that is easy for them to grasp. While most of the paper is well written, there are some sections that are not clear (especially section 2):
-	Terms such as LCF, OCaml-top level, deBruijn indices have been used without explaining or any references. These terms might be trivial in ITP literature, but were hard for me to follow.  
-	Section 2 describes how the data was splits into train and test set. One thing which is unclear is – can the examples in the train and test set be statements about the same conjecture or are they always statements about different conjectures? 


It also unclear how the deep learning models are applied. Let’s consider the leftmost architecture in Figure 1. Each character is embedded into 256-D vector – and processed until the global max-pooling layer. Does this layer take a max along each feature and across all characters in the input? 

My another concern is only deep learning methods are presented as baselines. It would be great to compare with standard NLP techniques such as Bag of Words followed by SVM. I am sure these would be outperformed by neural networks, but the numbers would give a sense of how easy/hard the current problem setup is. 

Did the authors look at the success and failure cases of the algorithm? Are there any insights that can be drawn from such analysis that can inform design of future models? 

Overall I think the research direction of using ML for theorem proving is an interesting one. However, I also feel the paper is quite opaque. Many parts of how the data is constructed is unclear (atleast to someone with little knowledge in ITPs). If authors can revise the text to make it clearer – it would be great. The baseline models seem to perform quite well, however there are no insights into what kind of ability the models are lacking. Authors mention that they are unable to perform logical reasoning – but that’s a very vague statement. Some examples of mistakes might help make the message clearer. Further, since I am not well versed with the ITP literature it’s not possible for me to judge how valuable is this dataset. From the references, it seems like it’s drawn from a set of benchmark conjectures/proofs used in the ITP community – so its possibly a good dataset. 

My current rating is a weak reject, but if the authors address my concerns I would change to an accept.

[Final Decision · Program Chairs · 06 Feb 2017]
**ICLR committee final decision**

The paper presents a new dataset and initial machine-learning results for an interesting problem, namely, higher-order logic theorem proving. This dataset is of great potential value in the development of deep-learning approaches for (mathematical) reasoning.
 
 As a personal side note: It would be great if the camera-ready version of the paper would provide somewhat more context on how the state-of-the-art approaches in automatic theorem proving perform on the conjectures in HolStep. Also, it would be good to clarify how the dataset makes sure there is no "overlap" between the training and test set: for instance, a typical proof of the Cauchy-Schwarz inequality employs the Pythagorean theorem: how can we be sure that we don't have Cauchy-Schwarz in the training set and Pythagoras in the test set?